# Diagnostic Capability and Improved Clinical Management of 18F-DCFPyL-PSMA PET/CT in Occult Biochemical Recurrence of Prostate Cancer After Prostatectomy

**DOI:** 10.3390/cancers17081272

**Published:** 2025-04-09

**Authors:** Francesco Amorelli, Palmira Foro, Juan Sebastian Blanco, Abrahams Ocanto, Augusto Natali, Lluis Fumado, Pedro Plaza

**Affiliations:** 1Department of Radiation Oncology, Hospital Del Mar, 08003 Barcelona, Spain; pforo@psmar.cat; 2PhD Program, Doctoral School, Pompeu Fabra University, 08002 Barcelona, Spain; pedrojose.plaza.lopez@psmar.cat; 3Department of Nuclear Medicine, Hospital Del Mar, 08003 Barcelona, Spain; juan.sebastian.blanco@psmar.cat; 4Department of Radiation Oncology, Hospital Universitario San Francisco de Asís, GenesisCare, 28002 Madrid, Spain; abrahams.ocanto@genesiscare.es; 5Department of Laboratory Medicine, Hospital Josep Trueta, 17007 Girona, Spain; aonatali.bcn.ics@gencat.cat; 6Department of Urology, Hospital Del Mar, 08003 Barcelona, Spain; lfumado@psmar.cat

**Keywords:** prostate cancer, occult biochemical recurrence, radical prostatectomy, 18F-DCFPyL PET/CT, detection rate

## Abstract

In the setting of biochemical recurrence of prostate cancer following primary radical treatment, conventional imaging studies such as computed tomography (CT Scan) and bone scintigraphy have been the standard methods of choice for years. However, these diagnostic techniques have limitations, particularly in detecting extra prostatic disease. PET-PSMA is a relatively new molecular imaging technique that, in recent years, has accumulated substantial evidence with various PSMA ligands, establishing itself as an effective tool for restaging and detecting distant disease in patients with biochemical recurrence of prostate cancer. The aim of this study is to assess the diagnostic performance of 18F-DCFPyL PSMA in patients with occult biochemical recurrence of prostate cancer following radical prostatectomy as primary treatment and to evaluate how positive or negative PET results can optimize the management of these patients.

## 1. Introduction

In patients with localized PC, BCR after RP occurs in 20% to 50% of cases [1,2]. BCR is defined by a PSA level exceeding 0.2 ng/mL [3]. Conventional imaging studies, such as contrast-enhanced CT Scans, pelvic magnetic resonance imaging (MRI), and bone scintigraphy, often yield false-negative results for the early detection of local and/or systemic recurrences, especially at low PSA levels, posing a significant challenge in determining diagnostic and therapeutic strategies [4,5,6]. As a result, the management of these patients is typically guided by international consensus guidelines [7,8,9], often involving salvage radiotherapy to the prostate bed [10], which achieves a 5-year local control rate of 56% [11,12]. However, treatment failure is frequently attributed to disease located outside the prostate bed [13,14].

Currently, there is no clear consensus on the optimal imaging modality for patients with BCR and low PSA levels. Conventional imaging techniques are less sensitive than PET/CT in detecting BCR of PC. PET/CT with various radiotracers has shown an improved detection of recurrent disease and is increasingly used in the diagnostic approach for PC patients with BCR, offering higher sensitivity at earlier stages [15]. Radiotracers such as 18F/11C-Choline and 18F-Fluciclovine have become well-established options in this scenario [16,17]; however, their use has declined as PET/CT with PSMA ligands offers superior sensitivity and specificity for BCR, particularly in cases of early recurrence with low PSA levels [18]. Consequently, PSMA labeled with 18F has become a focus of ongoing research due to its promising results and superior diagnostic performance in overcoming this limitation [19,20].

18F-DCFPyL, 18F-PSMA-1007, and 68Ga-PSMA-11 are considered a single class of radiotracers due to their similarity and are collectively referred to as PSMA ligands. All of them are radiotracers that bind with high affinity to the extracellular domain of PSMA and have demonstrated efficacy in detecting PC at various stages of the disease, with increased uptake correlating with tumor aggressiveness [21,22]. 18F-DCFPyL is a radiotracer that binds to the extracellular domain of PSMA with high affinity and has demonstrated efficacy in detecting PC [23]. It offers superior sensitivity and specificity for identifying local and/or distant disease in cases of BCR compared to conventional imaging techniques [24,25,26]. It has been assumed that the earlier localization of recurrence will result in better outcomes, although there is still little evidence for the effects of a change in management based on PSMA PET in patients with BCR. PSMA PET/CT imaging is recommended in international guidelines [27,28]. However, there remains no clear consensus on the optimal PSA threshold at which the test should be requested.

Our objective is to evaluate the diagnostic capability of 18F-DCFPyL PET/CT in patients with early BCR of PC following RP and to assess how positive findings from this imaging technique can influence the therapeutic decision-making and clinical management of these patients.

## 2. Materials and Methods

We performed a prospective observational single-center study in Barcelona, Spain. Between September 2020 and January 2023, we selected 85 patients diagnosed with PC presenting BCR after RP as primary treatment, with PSA levels between 0.2 and 2.0 ng/mL, who were being considered for salvage radiotherapy, as well as those with persistent detectable PSA levels despite prior adjuvant or salvage radiotherapy of the prostate bed. The selected patients with occult BCR showed negative findings on conventional imaging with contrast-enhanced CT Scans, pelvic MRI, and bone scintigraphy for local or distant disease.

We evaluated the diagnostic capability of 18F-DCFPyL PET/CT by analyzing the disease DR per patient in relation to clinical variables such as Gleason score and ISUP risk group, as well as biochemical variables, including PSA values (<0.5, 0.5–1, or >1 ng/mL) and DT-PSA (<6, 6–12, and >12 months). The total number of lesions diagnosed by 18F-DCFPyL PET/CT were classified into local recurrence, pelvic and extra-pelvic lymph node, bone, and visceral disease. Optimal cutoff points for PSA (ng/mL) and DT-PSA (in months) were determined using ROC curves. After the molecular imaging study, potential changes in therapeutic decisions and overall clinical management based on positive PSMA findings were assessed. During patient selection, clinical information was collected, including the type of surgery, histological characteristics of the primary tumor (initial Gleason score and ISUP classification), disease stage according to the TNM classification, use of androgen deprivation therapy (ADT), PSA values in ng/mL, and DT-PSA in the months prior to performing the PET/CT with 18F-DCFPyL. Clinical and therapeutic evaluations were conducted by expert radiation oncologists.

### 2.1. Imaging Protocol and Analysis

All patients underwent PET/CT with 18F-DCFPyL PSMA in the Nuclear Medicine Department of the Hospital del Mar in Barcelona, Spain. Patients received an intravenous injection of between 299 and 333 MBq of 18F-DCFPyL, synthesized under good manufacturing practices and supplied by Curium Pharma, Spain. Ninety minutes later, a whole-body PET scan with diagnostic CT was performed using a Siemens Biograph 40m CT tomograph, following the EANM Guidelines’ recommendations [29]. Iodinated intravenous contrast was administered unless contraindicated, and furosemide was not given. When necessary, an additional 5 min late pelvic PET scan was performed. The PET/CT with 18F-DCFPyL images were reviewed using SyngoVia-20 (Siemens Healthineers, Erlangen, Germany) by two experienced nuclear medicine physicians. Any disagreements in their reports were resolved by consensus. The standardized evaluation of images was based on the PSMA-Ligand PET/CT interpretation for Molecular Prostate Cancer (PROMISE and PROMISE V2) criteria [30,31]. Only 13 BIS clear foci of abnormal uptake not associated with physiological uptake and classified as consistent or suggestive by the PROMISE V2 assessment were considered positive. After performing the PET/CT with 18F-DCFPyL, the positive findings were classified into local recurrence, pelvic and extra-pelvic lymph node disease, bone disease, and visceral disease.

The nuclear medicine physicians employed had extensive experience in PET/CT studies for prostate cancer, utilizing 11C and 18F-choline ligands for over 10 years and three different PSMA ligands since 2018.

### 2.2. Statistical Analysis

Quantitative variables were described through means and standard deviation. Qualitative variables were described through a frequencies table (number and percentage). Between-group comparisons were performed for negative vs. positive results. The statistical tests used for these comparisons were the Chi-square or Fisher exact, as appropriate, for categorical variables and Mann–Whitney U for continuous variables. The choice of this non-parametric option was because of the violation of the normality assumption for a significant number of the continuous variables. Finally, multivariate logistic regression was performed to check factors associated with positive results. Variables in the multivariate analysis were introduced based on clinical criteria, considering those that were statistically significant or nearly significant (*p*, 0.05–0.1) in the bivariate analysis. Optimal cutoff points for PSA levels (ng/mL) and PSA-DT (months) were determined using ROC curves. Statistical analyses were performed using STATA 15.1. Results were considered as statistically significant at *p*-value < 0.05

## 3. Results

A total of 85 patients were eligible for this study. The patient characteristics, listed in Table 1, were as follows: age 48–78 years (mean age 69 years); initial staging at diagnosis as T2 (50.58%)–T3 (48.23%); N1 cases (4.7%); Gleason score < 8 (43.2%); and >8 (56.4%) with mean PSA level 11.24 ng/mL at diagnosis. The surgical approach performed was RP in all patients and pelvic lymph node dissection (PLND) in 51.76%; adjuvant radiotherapy was performed in 14.11%, salvage radiotherapy 38.82%, and ADT had been used in 38.82% of cases.

Positive results were obtained in 53% of patients (45/85) in the 18F-DCFPyL PET/CT study. The median PSA level before the scan was 0.59 ng/mL (95% CI, *p* = 0.004, 0.29–1.0 ng/mL), and the median DT-PSA was 7 months (95% CI, *p* = 0.005, 3–9 months). Among the positive results, 54% of the cases had a Gleason score ≥ 8, 28.9% were in ISUP risk group 3, 57.7% in groups 4–5, and 15.6% had pN1 involvement at the time of diagnosis, with a similar proportion of T2c-T3b cases in both groups.

### 3.1. 18F-DCFPyL PET/CT Detection Rate

The DRs were 31.3%, 60%, and 77.8% for PSA levels below 0.5 ng/mL, between 0.5 and 1.0 ng/mL, and above 1.0 ng/mL, respectively. Regarding DT-PSA, the DRs were 61.5%, 50%, and 26.7% for cases with less than 6 months, between 6 and 12 months, and more than 12 months, respectively, and 24%, 59%, and 68.4% for the ISUP risk groups 1–2, 3, and 4–5, respectively. All results were statistically significant, with a *p*-value of <0.001 for the PSA ng/mL, <0.005 for DT-PSA (months), and <0.02 for the ISUP risk groups (Figure 1).

In our cohort, among the positive 18F-DCFPyL PET/CT results, a total of 90 lesions were identified (21 in the prostate bed, 48 pathological lymph nodes, 18 bone lesions, and 3 visceral lesions) that were not detected by conventional imaging (Figure 2, Figure 3 and Figure 4).

The DR by location was 22.2% for local disease in the prostate bed (10 patients), 51.1% for lymph node involvement (23 patients), 20% for bone lesions (9 patients), and 6.7% for visceral involvement (3 patients) (Figure 5).

Additionally, positive findings by location were analyzed in relation to PSA levels (ng/mL) and DT-PSA in months. The largest group of patients with positive PET/CT had PSA levels between 0.5 and 1.0 ng/mL and a DT-PSA time of less than 6 months (Figure 6).

ROC analysis showed a PSA value of 0.55 ng/mL as the optimal cutoff value for predicting positive PET/TC with 18F-DCFPyL (sensitivity 84%; specificity 60%), with an area under the curve (AUC) of 0.72 (95% CI 0.51–0.85), and a DT-PSA of 9.2 months (sensitivity 89%; specificity 37%), with an AUC of 0.60 (95% CI 1.65–14.3) (Figure 7).

### 3.2. Changes in Therapeutic Decisions and Overall Clinical Management

In patients with positive 18F-DCFPyL PET/CT findings, therapeutic strategies were optimized in 84.4% (38/45) of cases (*p* < 0.001). Initially, the 85 patients with BCR of PC included in this cohort were being considered for salvage or adjuvant radiotherapy to the prostate bed, ADT for those with prior prostate bed irradiation, or an expectant management approach. Based on PET/CT results, therapeutic decisions and overall clinical management were adjusted according to the positive or negative findings. In 38 patients with positive results, significant changes in local therapeutic management were implemented, including recommendations for radiotherapy to pelvic lymph node areas (PLNs) with or without dose escalation to target affected lymph nodes, stereotactic body radiotherapy (SBRT) guided by PSMA-positive metabolic volume, and prostate bed radiotherapy with dose escalation in cases of local recurrences showing PSMA uptake. Furthermore, due to the proportion of extra-pelvic lymph node, bone, and visceral lesions detected by the PSMA molecular study, 42.3% of patients were indicated for systemic treatment with androgen receptor signaling inhibitors (ARSIs).

Among the negative results group (40/85), 41% of patients followed the initial recommendation of radiotherapy to the prostate bed, while 59% opted for expectant management with clinical follow-up (Figure 8).

After implementing the changes in therapeutic approach based on PSMA results and following a 24-month follow-up period, the median PSA level was 0.08 ng/mL. Among PSMA-positive patients, a PSA reduction of more than 50% was observed in 65.8%, a reduction of less than 50% in 26.3%, and no PSA response in 7.9%.

## 4. Discussion

The PSMA ligands used in PET/CT are a class of radiotracers designed to target the prostate-specific membrane antigen. The first PSMA ligand was radio synthesized and validated in preclinical models at Johns Hopkins University [32]. Later, 68Ga-PSMA-11 was developed by the Heidelberg group [33]. PSMA ligands differ in terms of the radioisotope used, their underlying radiochemistry, and their biodistribution. Differences in physiological distribution and challenges in image interpretation are well known. However, there is currently no evidence that any specific PSMA radioligand provides superior clinical outcomes over others [22].

Among PSMA ligands, 68Ga-PSMA-11 has the longest history of use and the strongest scientific evidence supporting its efficacy. Currently, 68Ga-PSMA-11, 18F-DCFPyL, and 18F-PSMA-1007 are the most widely used in clinical trials. All of them bind with high affinity to the extracellular domain of PSMA and have demonstrated efficacy in detecting PC at various stages, with increased uptake correlating with tumor aggressiveness. 18F-DCFPyL has shown a similar lesion detection rate to 68Ga-PSMA-11, without an increase in false positive rates. Additionally, 18F-DCFPyL has demonstrated a higher detection rate compared to 68Ga-PSMA-11, which may be attributed to the superior spatial resolution of 18F imaging [21]. Due to their similarities, 68Ga-PSMA-11, 18F-DCFPyL, and 18F-PSMA-1007 are considered a single class of radiotracers and are collectively referred to as PSMA ligands [22].

Conventional imaging techniques such as ultrasound, CT, and MRI are less sensitive than PET imaging in detecting BCR of PC. PET radiotracers such as 18F/11C-Choline and 18F-Fluciclovine were once well-established options for this clinical scenario. However, their use has declined because PET with PSMA ligands labeled with 18F (e.g., 18F-DCFPyL) or 68Ga (e.g., 68Ga-PSMA-11) offers superior sensitivity and specificity for identifying local and distant disease in cases of BCR, particularly in early recurrence with low PSA levels [15,18,19,25]. Furthermore, even in cases of incidental PC diagnosis, PSMA-PET has proven to be significantly more sensitive and specific than CT, MRI, and bone scintigraphy for staging lymph node and bone metastases. It is also more sensitive than multiparametric MRI for local tumor staging when PSMA-PET/MRI is used [34].

International guidelines recommend performing PSMA PET/CT imaging for BCR of PC. However, there is still no clear consensus on the optimal PSA threshold at which the test should be ordered [28].

Our prospective observational study demonstrates the diagnostic capability and clinical utility of 18F-DCFPyL PET/CT in patients with early occult BCR of PC following RP. In these patients, molecular imaging facilitates the early localization of recurrent disease and enables treatment optimization. The current EAU clinical guidelines recommend performing PSMA-targeted PET/CT in patients with BCR after RP, particularly with PSA levels around 0.2–0.5 ng/mL, provided it influences subsequent therapeutic decisions [27,28]. Our findings demonstrate a robust disease DR across varying PSA levels and DT-PSAs, confirming superior diagnostic performance when compared to conventional imaging techniques, including contrast-enhanced CT Scans, pelvic MRI, and bone scintigraphy, for identifying local or distant disease. There is a positive correlation between PSA levels (ng/mL), DT-PSA (months), and higher-risk ISUP groups with the ability of 18F-DCFPyL PET/CT to detect disease (AUC 0.74). The diagnostic power of the test increases significantly with higher PSA levels and shorter DT-PSAs.

Most patients in our study were high-risk cases (Gleason ≥ 8, ISUP 4–5) with pre-PET/CT PSA values between 0.5 and 1.0 ng/mL achieving a 60% DR for recurrent disease. Among patients with PSA < 0.5 or >1.0 ng/mL, the DRs were 31.3% and 77.8%, respectively, highlighting the reliability of this imaging modality even at low PSA levels, overcoming a critical limitation of conventional imaging techniques. Regarding DT-PSA, we observed a higher distribution of patients with a DT-PSA < 6 months, followed by those with a TD-PSA between 6 and 12 months, with RDs of 61.5% and 50%, respectively. These results are consistent with findings from pivotal studies and reinforce the critical role of PSMA PET/CT in the management of BCR of PC. The DR of disease was further analyzed in studies such as the CONDOR trial (Morris et al.) [35], a Phase 3 study involving 208 patients with negative conventional imaging. 18F-DCFPyL PET/CT achieved significant DRs even at low PSA levels, with DRs ranging from 36.2% for PSA < 0.5 ng/mL to 96.7% for PSA ≥ 5 ng/mL. Remarkably, 63.9% of patients experienced changes in clinical management. Similar findings by Fendler et al. [36] and Aydin et al. [37] highlight the impact of PSMA PET/CT in the early detection of recurrent disease. Fendler reported DRs of up to 80% in patients with rising PSA, revealing metastases that were unidentifiable by conventional imaging. Likewise, Aydin demonstrated the ability of PSMA PET/CT to detect metastases outside standard radiotherapy fields in patients with PSA ≤ 1 ng/mL, further underscoring its value in postoperative decision-making. Hoffman et al. [38] further explored the relationship between PSA levels, kinetics, and lesion detection using 68Ga-PSMA-11 PET/CT in 581 patients with BCR. Their DRs ranged from 40% for PSA levels between 0.2 and 0.5 ng/mL to 94% for PSA > 5 ng/mL, showing a positive correlation for disease detection with high PSA and short DT-PSA values. These findings emphasize the importance of integrating PSA metrics into decision-making for PSMA PET/CT imaging [39,40]. Systematic reviews and meta-analyses, such as that conducted by Treglia et al. [41], corroborated these observations. Their analysis of six studies (645 patients) reported a pooled DR of 86% for PSA ≥ 0.5 ng/mL and 49% for PSA < 0.5 ng/mL, confirming a significant association between PSA levels and DR. Furthermore, large-scale multi-center studies, such as that by Afshar-Oromieh et al. [42] (2533 patients), confirmed a progressive increase in DR from 43% for PSA ≤ 0.2 ng/mL to 93% for PSA > 10 ng/mL. These findings, along with our own data, reinforce the crucial role of molecular imaging with PSMA in bridging the gap between clinical suspicion and disease localization [36,43], particularly in high-risk or rapidly progressive disease scenarios.

In the study by Hoffman et al. [38] a PSA threshold of 1.24 ng/mL was identified as the optimal cutoff for predicting positive scan results. Similarly, our ROC curve analysis yielded favorable findings, establishing an optimal PSA cutoff of 0.55 ng/mL and a DT-PSA cutoff of 9.2 months for predicting positive PET/CT outcomes.

In our cohort, 90 PSMA-positive lesions were identified that were undetectable by conventional imaging techniques. These included local recurrences in the prostate bed (22.2%), pathological pelvic and extra-pelvic lymph nodes (51.1%), bone lesions (20%), and visceral metastases (6.7%), highlighting the ability of this imaging modality to map disease and guide therapy based on metabolic imaging. These findings align with similar results reported in the current literature [44,45,46]. The OSPREY trial by Pienta and Gorin et al. [25] is one of the most pivotal studies on the use of 18F-DCFPyL PET/CT in PC patients. This Phase 2/3 clinical trial evaluated the diagnostic accuracy of 18F-DCFPyL PET/CT in high-risk PC patients and those with BCR. The study demonstrated the ability of 18F-DCFPyL PET/CT to detect previously unsuspected disease in the prostate bed, lymph nodes, and distant metastases, enabling more precise staging and better-informed therapeutic decisions. Notably, the modality exhibited high sensitivity (89%) for lesion detection in patients with low PSA levels, underscoring its clinical utility. These findings suggest that molecular imaging with 18F-DCFPyL PET/CT provides substantial clinical benefits in the early detection of recurrent disease and the development of improved treatment strategies. Similarly, Roach et al. [47] conducted a prospective multi-center study involving 431 prostate cancer patients across four Australian centers, which demonstrated the significant clinical impact of 68Ga-PSMA PET/CT. Molecular imaging led to changes in planned clinical management in 51% of cases, with a notably higher impact in patients with BCR (62%) compared to those undergoing primary staging (21%). The scans revealed previously undetected disease in the prostate bed (27%), locoregional lymph nodes (39%), and distant metastases (16%), highlighting the superior diagnostic capabilities of 68Ga-PSMA PET/CT.

PSMA PET can also be used to optimize salvage treatment. Emmett et al. [48] studied 164 men considered eligible for salvage radiotherapy due a rising PSA, after RP, to PSA readings between 0.05 and 1.0 ng/mL. PSMA PET was independently predictive of treatment response to salvage radiotherapy. A negative or fossa-confined PSMA PET result predicted a high response to salvage fossa radiotherapy. On the other hand, nodes or distant disease in PET PSMA related to a poor response to radiotherapy. These findings align closely with our analysis, underscoring the important role of PSMA PET/CT in uncovering occult disease and enhancing treatment decision-making [49,50]. We observed the significant impact of molecular imaging with 18F-DCFPyL on therapeutic decision-making, consistent with findings from previous studies. Among patients with positive results, therapeutic strategies were optimized in 84.4% of cases. This included adjustments in radiotherapy planning as well as a higher initiation rate of systemic therapies involving androgen receptor signaling inhibitors (ARSIs) combined with ADT. These changes in therapeutic decision-making highlight the potential of PSMA PET/CT to redefine disease staging, tailor treatments, and enhance overall clinical management.

In summary, our findings highlight the superior diagnostic capability of 18F-DCFPyL PET/CT over conventional imaging, demonstrating significantly higher sensitivity in detecting recurrent lesions in patients with BCR of PC. Additionally, our results underscore the clinical value of 18F-DCFPyL PET/CT in managing BCR of PC, as it enhances diagnostic accuracy and facilitates the development of personalized treatment strategies, offering both localized and systemic therapeutic options.

### Limitations and Future Directions

There are some limitations in our study. We started with a prospective observational study involving a relatively small sample, with the objective of assessing the diagnostic capacity of 18F-DCFPyL PET/CT in patients with BCR of PC. Although it is not the main objective of our study, one of the primary limitations is that, currently, we do not have long-term outcome data to evaluate overall survival (OS) and progression-free survival (PFS). Therefore, a second analysis could be considered for a future publication, where we assess these patients with extended follow-ups and include PFS and OS data. We are currently working on future publications, including a review of different 18F-labeled PSMA ligands, directly comparing them with conventional imaging techniques. In our current results, no comparative analysis between PET-PSMA and conventional imaging techniques was performed, as the patients selected for the study had no visible disease on conventional imaging and the aim was to determine whether they had disease detectable by metabolic imaging with PET-PSMA. In the future, we plan to expand our sample size for future publications and are working continuously to enlarge our study cohort, focusing on patients with low PSA levels in the context of BCR. Additionally, we are working on detailed comparison analyses focused on radiotherapy fields adapted by PSMA-positive metabolic imaging versus conventional imaging-guided treatment fields.

## 5. Conclusions

18F-DCFPyL PET/CT is an effective and reliable tool for the early diagnosis of patients with PC experiencing occult BCR after primary treatment with RP, showing favorable disease DRs even with low PSA values. In our study, this led to a significant impact on decision-making and treatment optimization in 84.4% of patients with positive PSMA results. We strongly believe that molecular imaging with PSMA is crucial for accurate diagnosis and therapeutic management in patients with BCR of PC.

## Figures and Tables

**Figure 1 cancers-17-01272-f001:**
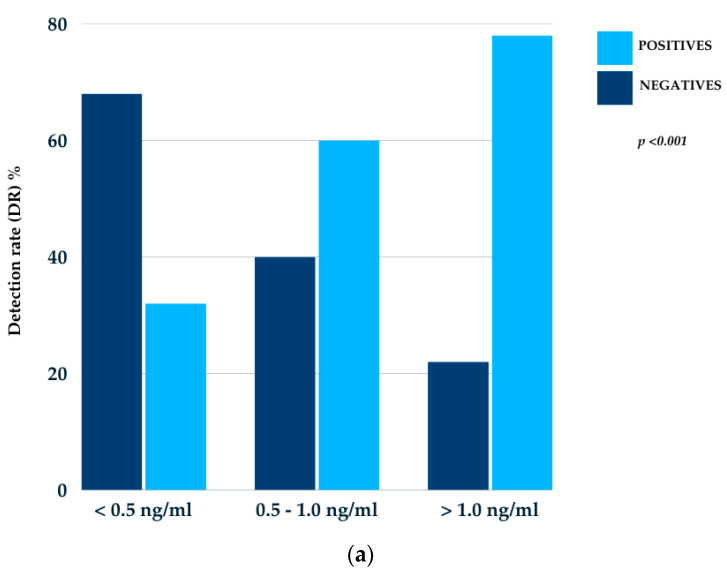
Disease detection rates of 18F-DCFPyL PET/TC in relation to PSA ng/mL (**a**), DT-PSA in months (**b**), and ISUP risk group (**c**).

**Figure 2 cancers-17-01272-f002:**
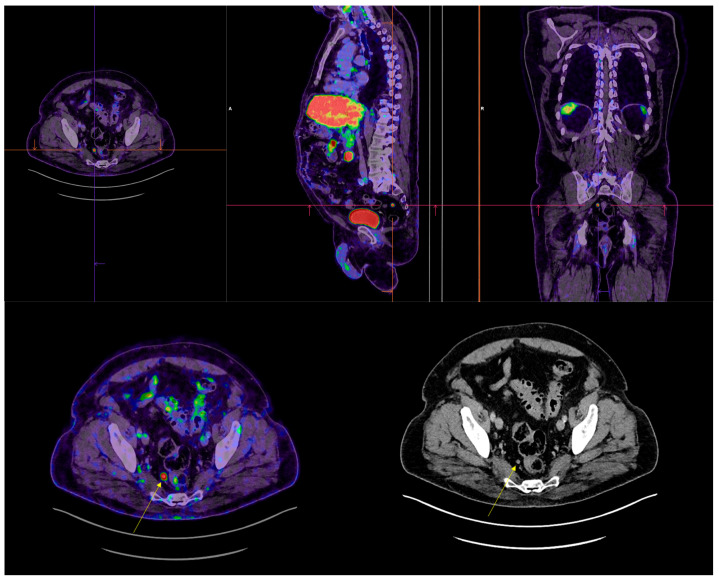
Pararectal metastases in lymph nodes (indicated by yellow arrow) with PSA 0.47 ng/mL and DT-PSA 8 months. Not visualized in conventional imaging study.

**Figure 3 cancers-17-01272-f003:**
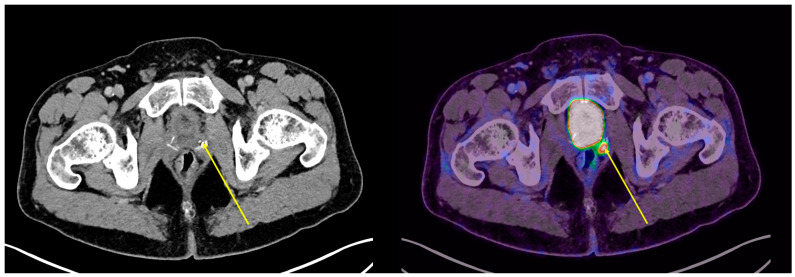
Local recurrence with no radiological findings (indicated by yellow arrow). PSA: 0.28 ng/mL; DT-PSA: 8.5 months.

**Figure 4 cancers-17-01272-f004:**
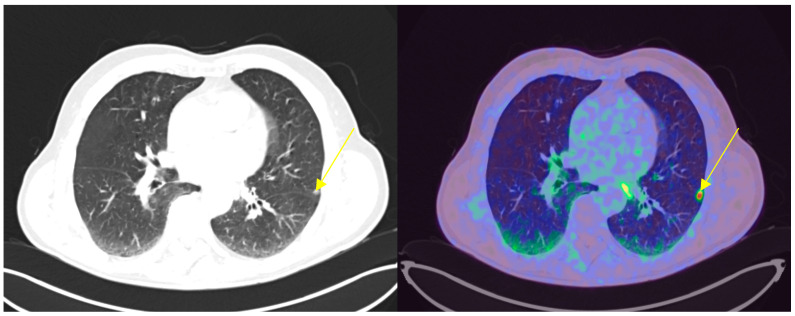
Lung metastases (indicated by yellow arrow). Radiological finding alone was non-specific. PSA: 0.77 ng/mL; DT-PSA: 6 months.

**Figure 5 cancers-17-01272-f005:**
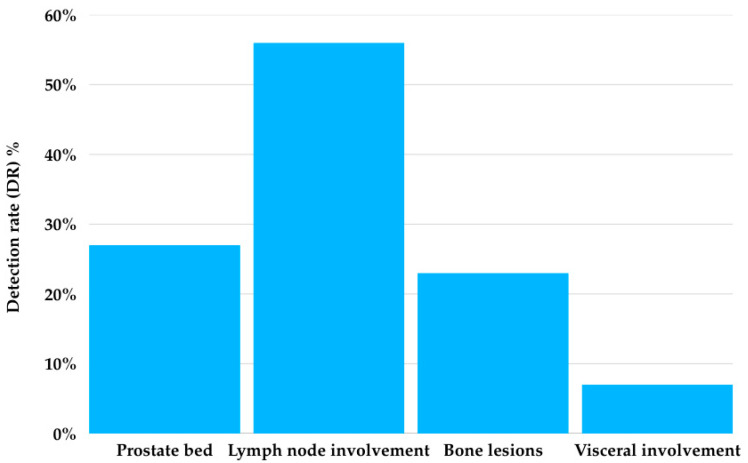
Positive patient detection rate of 18F-DCFPyL PET/CT by location.

**Figure 6 cancers-17-01272-f006:**
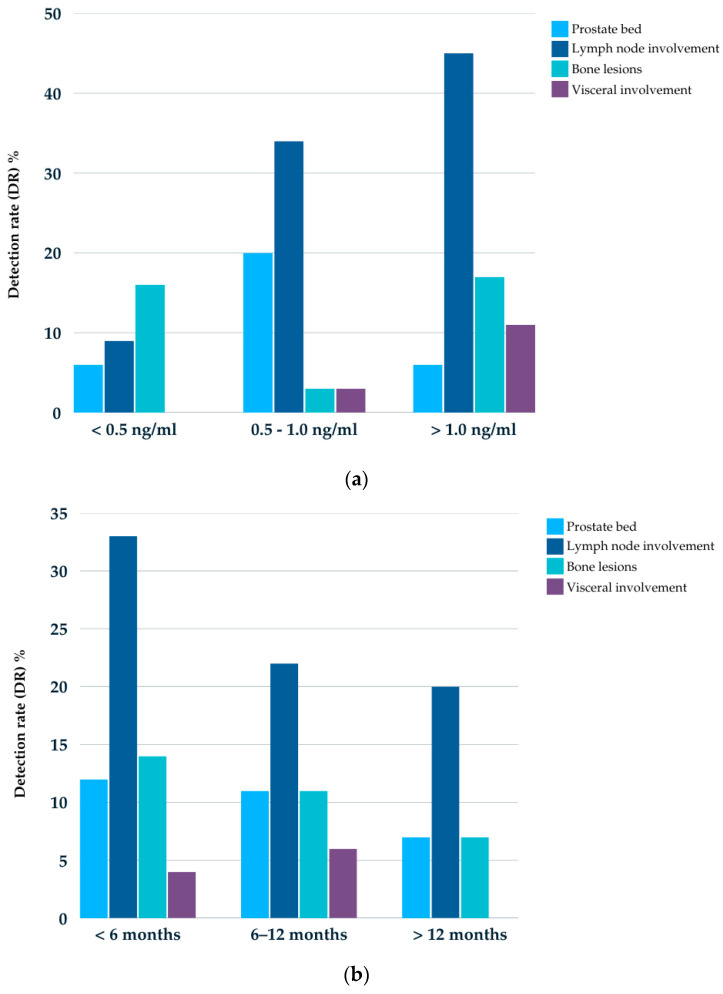
(**a**) Detection rate 18F-DCFPyL PET/TC in relation to PSA ng/mL values by location, (**b**) Detection rate 18F-DCFPyL PET/TC in relation to DT-PSA in months by location.

**Figure 7 cancers-17-01272-f007:**
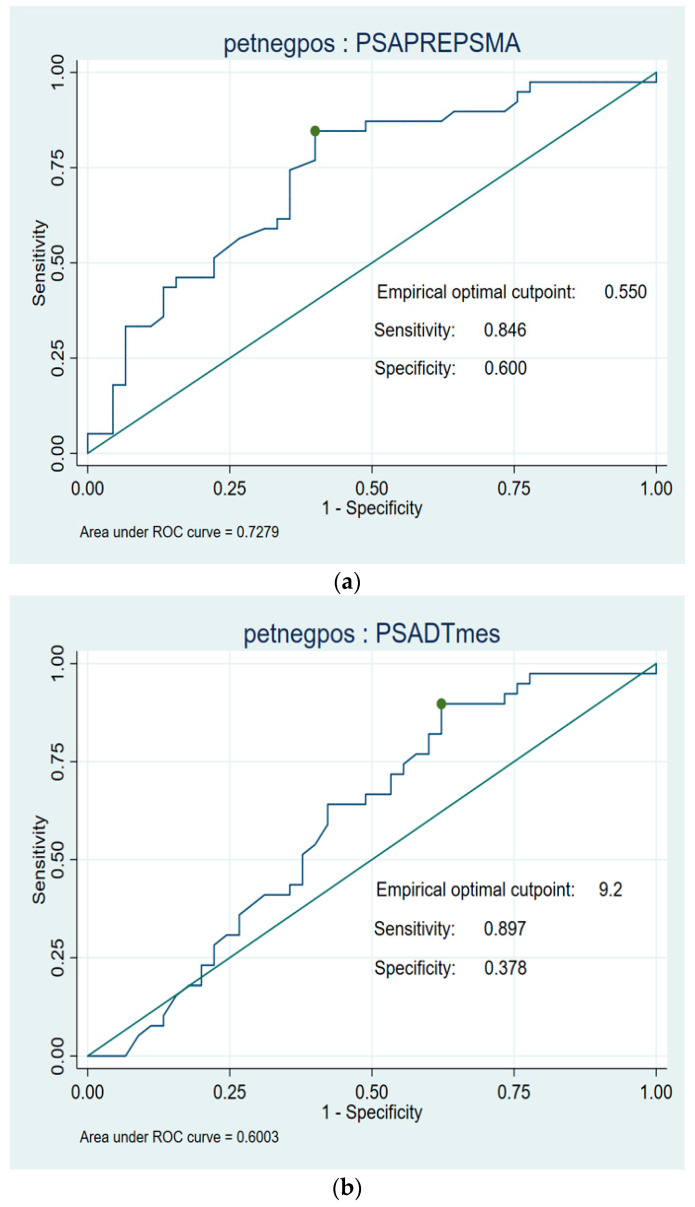
(**a**) Optimal cutoff point RoC curves for PSA (ng/mL). (**b**) Optimal cutoff point RoC curves for DT-PSA (months).

**Figure 8 cancers-17-01272-f008:**
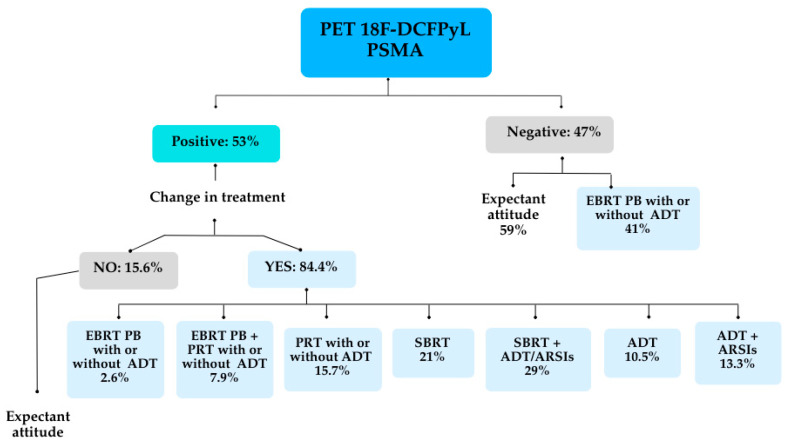
Impact of PET/TC with 18F-DCFPyL on clinical management based on positive/negative findings.

**Table 1 cancers-17-01272-t001:** Population characteristics (n = 85).

Characteristic	Value
Age	48–74 years
Median age	69 years
Mean diagnosis PSA	11.24 ng/mL
**TNM**	
T2 a-b	7
T2c	36
T3a	21
T3b	20
T4	1
Nx	32
N0	49
N1	4
Mx	32
M0	53
**Histological characteristics**	
Gleason score < 8	37
Gleason score > 8	48
ISUP 1–2	25
ISUP 3	22
ISUP 4–5	38
**Primary treatment**	
PR	41
PR + PLND	44
**Radiotherapy**	
NO	40
Adjuvant	12
Salvage	33
**ADT**	
Yes	33
No	52
**Median PSA** **(pre-DCFPyL PET/CT)**	**0.59 ng/mL** **(0.29–1.0)**
**Mean DT-PSA** **(pre-DCFPyL PET/CT)**	**7.2 months** **(3–9.1)**

## Data Availability

The original contributions presented in this study are included in the article. Further inquiries can be directed to the corresponding author.

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
