# Peer review of "Diagnostic Capability and Improved Clinical Management of 18F-DCFPyL-PSMA PET/CT in Occult Biochemical Recurrence of Prostate Cancer After Prostatectomy"

_cancers, 2025, doi:10.3390/cancers17081272_

Round 1
Reviewer 1 Report
Comments and Suggestions for Authors
This is a well performed study reporting the experience of the authors in performing 18F-DCFPyL-PSMA PET/CT in occult biochemical recurrence of prostate cancer followimg prostatectomy.
The manuscript is well written and the results are clearly presented. I would sugggest that the authors include in the discussion the potential differences (i.e: advantages and disadvantages, from the perspective of results, costs, easiness to handle the trazer, etc) between this type of PET and Gallium-PSMA, if any.
Thanks for the opportunity to review this paper.
Author Response
Comments and Suggestions for Authors:
This is a well performed study reporting the experience of the authors in performing 18F-DCFPyL-PSMA PET/CT in occult biochemical recurrence of prostate cancer following prostatectomy.
Comment 1: The manuscript is well written, and the results are clearly presented. I would suggest that the authors include in the discussion the potential differences between this type of PET and Gallium-PSMA, if any.
Response and revisions:
Dear Reviewer, thank you very much for your suggestion. We have added a paragraph in the introduction (L 77-81) discussing the different PSMA tracers and have also included a comprehensive description of their advantages and clinical applications, supported by updated references, in the discussion section of our final manuscript (L 353-385).
I attach a word document to this answer with a detailed summary on the use of PSMA ligands.
Once again, on behalf of our entire team, we sincerely appreciate your comments.

Reviewer 2 Report
Comments and Suggestions for Authors
L 80 you comment that little evidence is on outcome of restaging PSMA PET/CT for BCR. You report 45 patients had positive restaging PSMA PET. With a minimum follow-up of 24 months you could add information on the outcome of PSA follow-up: How many patients had at least a 50% PSA decline end 2024 etc. L 113 I assume that most patients were tested at Dept of Nucl med in Hospital del Mar Barcelona. L124 the year long experience of number of nuclear medicine physicians should be mentioned. The criteria for a positive site should be stated. You found PSA and PSA DT predicted positive PET results. Perhaps also high ISUP was associated with positive results? PSA and PSA DT may be combined to highest DR for PSA >1 & PSADT <6 months, and lowest DR for those with PSA <0.5 and PSADT >12 months. You may define 3-4 different risk groups. You may use logistic regression analysis You might add information on BCR-free survival for 40 PET negative patients and 45 PET positive patients. You expect the negative patients have better outcome than the positive patients You might add patients with PSA persistence. Reference list did not include [1, 2]
bibliograph
[1] le Guevelou J, Achard V, Mainta I, Zaidi H, Garibotto V, Latorzeff I, et al. PET/CT-Based Salvage Radiotherapy for Recurrent Prostate Cancer After Radical Prostatectomy: Impact on Treatment Management and Future Directions. Front Oncol 2021;11:742093.
[2] Cornford P, van den Bergh RCN, Briers E, Van den Broeck T, Brunckhorst O, Darraugh J, et al. EAU-EANM-ESTRO-ESUR-ISUP-SIOG Guidelines on Prostate Cancer-2024 Update. Part I: Screening, Diagnosis, and Local Treatment with Curative Intent. Eur Urol 2024.
Author Response
Dear Reviewer, first and foremost, I would like to express my sincerest gratitude for taking the time to review our work and provide your valuable suggestions and comments. Your feedback is greatly appreciated and has undoubtedly contributed to enhancing our manuscript.
I will address each of your comments point by point.
Comments and Suggestions for Authors
Point 1: L 80 you comment that little evidence is on outcome of restaging PSMA PET/CT for BCR. You report 45 patients had positive restaging PSMA PET. With a minimum follow-up of 24 months, you could add information on the outcome of PSA follow-up: How many patients had at least a 50% PSA decline end 2024.
Response and revisions:
Dear Reviewer, regarding point 1, although our primary objective was not to evaluate the biochemical response of the patients, your insightful suggestion prompted us to revisit these data. Upon reviewing our database, we recognized the relevance of incorporating your proposal into our results. Specifically, after implementing therapeutic adjustments based on PSMA-positive findings and following a 24-month follow-up period, we found that the median PSA level was 0.08 ng/mL. Among PSMA-positive patients, a PSA reduction of more than 50% was observed in 65.8%, a reduction of less than 50% in 26.3%, and no PSA response in 7.9%.
This revision has been incorporated into our results section, which you can find in the revised manuscript (Lines 346–350).
Point 2: L 113 I assume that most patients were tested at Dept of Nucl med in Hospital del Mar Barcelona. L124 the year long experience of number of nuclear medicine physicians should be mentioned.
Response and revisions:
Dear Reviewer, in response to point 2 of your observations, thank you for your comment. Indeed, all patients underwent the study with 18F-DCFPyL at Hospital del Mar in Barcelona, Spain. The nuclear medicine physicians at our center have extensive experience in PET/CT studies for prostate cancer, utilizing 11C and 18F-choline ligands for over 10 years and three different PSMA ligands since 2018.
This revision has been incorporated into the revised manuscript in the Materials and Methods section, under the imaging protocol. (Lines 135–137).
Point 3: You found PSA and PSA DT predicted positive PET results. Perhaps also high ISUP was associated with positive results?
Response and revisions:
Regarding point 3, we would like to express our gratitude once again for your observation and suggestion regarding our work. During the data collection and processing for the statistical analyses, we considered the ISUP risk group; however, it is true that during the writing of the manuscript, we focused more on PSA ng/mL and DT-PSA. Nevertheless, upon drafting our response to this point, we realized how valuable and enriching it has been to include this information in our results. We have made some additions in the revised manuscript.
L 167-168: we considered the ISUP risk group when describing the relevant variables for patients with positive PSMA studies. Among the total positives, 28.9% corresponded to ISUP 3 patients, while 57.7% were ISUP 4-5 patients.
L 175-176 and figure 1C: We included the data on the detection rates by ISUP risk group, which are 24%, 59%, and 68.4% for ISUP risk groups 1-2, 3, and 4-5, respectively.
Finally, during the calculations of the optimal cutoff points to predict a positive outcome, we also performed calculations considering the ISUP value. However, we did not include this value in the manuscript as we focused on PSA and DT-PSA for establishing the cutoff points.
I have attached the ROC curve for the ISUP risk group in the word document, which is not included in the final manuscript. If you believe this information can provide additional value beyond the data we have already included, we would be more than willing and happy to add it.
Point 4
a) PSA and PSA DT may be combined to highest DR for PSA >1 & PSADT <6 months, and lowest DR for those with PSA <0.5 and PSADT >12 months.
Response and revisions: regarding this question, it is undoubtedly a very interesting suggestion. During the study design, we considered analyzing disease detection rates for the variables PSA (ng/mL) and DT-PSA (months) both separately and together. However, we ultimately decided to conduct the analysis separately to align with the prevailing approach in clinical guidelines.
Although we did not perform a combined detection rate analysis in this study, we are currently working on two articles—one focused on 1007-PSMA and another comparing different PSMA tracers—where we aim to incorporate a joint analysis of PSA and DT-PSA in detection rates.
b) You may define 3-4 different risk groups. You may use logistic regression analysis You might add information on BCR-free survival for 40 PET negative patients and 45 PET positive patients.
Response and revisions: this idea has been highly enriching for our work. We consulted with the team of statisticians within our research group, and they indicated that our sample size is insufficient to perform a logistic regression analysis. However, given how interesting we found your suggestion, we have decided to incorporate it into our next two articles mentioned earlier. For these studies, we will have a larger sample size available for analysis.
Point 5: reference list did not include [1, 2]
Response and revisions: Dear Reviewer, I have reviewed the bibliographic citations. Thank you very much.
Once again, on behalf of our entire team, we sincerely appreciate your comments.

Reviewer 3 Report
Comments and Suggestions for Authors
The paper is interesting. A radical shift in the treatment of BR is ongoing thanks to PET PSMA. It is able to reveal metastasis that are occult to the conventional imaging. There are two issues that should be discussed by Authors 1) even if false positive rate is low, it may lead to a wrong treatment 2) PET PSMA based on 18F or Gallium have different outcomes and should not be considered the same imaging modality. That said, there are some other minor issues
1) Was the trial registered?
2) Was it approved by an Ethical Committee?
3) How was determined the dimension of the study population?
Author Response
Dear Reviewer, first and foremost, I would like to express my sincerest gratitude for taking the time to review our work and provide your valuable suggestions and comments. Your feedback is greatly appreciated and has undoubtedly contributed to enhancing our manuscript.
I will address each of your comments point by point.
Comments and Suggestions for Authors
The paper is interesting. A radical shift in the treatment of BR is ongoing thanks to PET PSMA. It is able to reveal metastasis that are occult to the conventional imaging.
Point 1: there are two issues that should be discussed by Authors
a) Even if false positive rate is low, it may lead to a wrong treatment
Response and revisions:
Dear Reviewer, regarding the first point, it is an interesting question and remains a challenge in clinical practice when evaluating changes in the therapeutic approach for patients based on the positive or negative results of PET-PSMA. As you mentioned, if we consult the most relevant literature (most of which is cited in our manuscript), the rate of false negatives is low, typically ranging from 5% to 15%, although some smaller publications even report rates as high as 20%. This presents a challenge when making decisions. One key aspect in addressing this gray area is the experience of nuclear medicine physicians, who work together to resolve uncertainties following international guidelines (for example, PROMISE V1 and V2).
Focusing on our results in the imaging protocol section of the materials and methods, we mention the experience of our team of nuclear medicine physicians, which comprises over 20 years of expertise. Any uncertainties that arose during the image assessment process were resolved by consensus, classifying them as positive or negative. From our perspective as radiation oncologists, the therapeutic approach guided by PSMA was implemented in 84.4% of patients with positive results, achieving a mean PSA level of 0.08 ng/ml after treatment. In the negative group, 41% followed the standard recommendation for radiotherapy of the prostate bed, with or without androgen deprivation therapy, resulting in a biochemical response of PSA >50%. Although the primary objective of our study was not to assess biochemical response, the results indicated that PSMA-guided treatment translated into a response for most patients.
However, this issue of false positives and negatives has motivated me to conduct further reviews. We are currently writing another article based on PSMA-1007 and are analyzing a comparison of PSMA ligands, where we are including biochemical response and progression-free survival in our patients, which we hope to publish soon.
b) PET PSMA based on 18F or Gallium have different outcomes and should not be considered the same imaging modality. That said, there are some other minor issues
Response and revisions:
Regarding this point, you are absolutely right, there is evidence in the literature comparing PET/CT with different PSMA ligands. PSMA ligands differ in terms of the radioisotope used, the underlying radiochemistry, and their biodistribution. Differences in physiological distribution and challenges in image interpretation are well known. However, there is currently no evidence that any specific PSMA radioligand provides superior clinical outcomes over others. Among PSMA ligands, 68Ga-PSMA-11 has the longest history of use and the strongest scientific evidence supporting its efficacy. Currently, 68Ga-PSMA-11, 18F-DCFPyL, and 18F-PSMA-1007 are the most widely used in clinical trials. All of them bind with high affinity to the extracellular domain of PSMA and have demonstrated efficacy in detecting prostate cancer at various stages, with increased uptake correlating with tumor aggressiveness. 18F-DCFPyL has shown a similar lesion detection rate to 68Ga-PSMA-11, without an increase in false positive rates. Additionally, 18F-DCFPyL has demonstrated a higher detection rate compared to 68Ga-PSMA-11, which may be at-tributed to the superior spatial resolution of 18F imaging. Due to their similarities, 68Ga-PSMA-11, 18F-DCFPyL, and 18F-PSMA-1007 are considered a single class of radiotracers and are collectively referred to as PSMA ligands.
Conventional diagnostic techniques such as ultrasound, CT, and MRI are less sensitive than PET imaging in detecting BCR of PC. PET radiotracers such as 18F/11C-Choline and 18F-Fluciclovine were once well-established options for this clinical scenario. However, their use has declined because PET with PSMA ligands labeled with 18F or 68Ga offers superior sensitivity and specificity for identifying local and distant disease in cases of BCR, particularly in early recurrence with low PSA levels. Furthermore, even in cases of incidental PC diagnosis, PSMA-PET has proven to be significantly more sensitive and specific than CT, MRI, and bone scintigraphy for staging lymph node and bone metastases.
We have added a paragraph in the introduction (L 77-81) discussing the different PSMA tracers and have also included a comprehensive description of their advantages and clinical applications, supported by updated references, in the discussion section of our final manuscript (L 353-385).
Point 2: there are some other minor issues
¿Was the trial registered? ¿Was it approved by an Ethical Committee?
Response and revisions: dear reviewer Thank you very much for your inquiry. Regarding point two, yes, our study was approved by the Clinical Research Ethics Committee of Hospital del Mar (CEIC-PSMAR) under approval number 2025/11956.
¿How was determined the dimension of the study population?
Response and revisions: as for the study population size, the sample size calculation was performed to demonstrate that the PET-PSMA complementary test alters the diagnostic approach in 80% of patients, with a 95% confidence level and a 10% margin of error. The required sample size (n) was approximately 62 patients.
Once again, on behalf of our entire team, we sincerely appreciate your comments.
Reviewer 4 Report
Comments and Suggestions for Authors
This study evaluates the diagnostic performance and clinical impact of 18F-DCFPyL PSMA PET/CT in patients with occult biochemical recurrence of prostate cancer following radical prostatectomy. Given the well-documented limitations of conventional imaging modalities in detecting early recurrence, this work aims to determine how PSMA-based molecular imaging informs therapeutic decision-making.
- While the study focuses exclusively on 18F-DCFPyL, it does not provide a direct comparison with other PSMA tracers (e.g., 68Ga-PSMA, 18F-Fluciclovine, 11C-Choline PET). Given that multiple PET tracers are available, a clear justification for selecting 18F-DCFPyL over others is necessary. The authors should dedicate a section explicitly discussing the rationale behind this choice, addressing differences in diagnostic accuracy, availability, half-life, resolution, and clinical relevance.
- Although the discussion briefly acknowledges certain limitations, a dedicated "Limitations and Future Directions" section would provide a clearer understanding for interpreting the study's findings. This section should address:
- The lack of long-term clinical outcome data (e.g., progression-free survival, overall survival).
- The absence of comparative analysis between 18F-DCFPyL PET/CT and conventional imaging modalities.
- The manuscript presents p-values to indicate statistical significance, but the graphical data should be revised for clarity. Specifically:
- Figures should explicitly indicate which groups show statistically significant differences rather than relying solely on p-values in the text.
- Box plots, confidence intervals, or annotations should be used to improve data interpretation.
- One of the study's key objectives is to evaluate how positive or negative 18F-DCFPyL PET/CT results optimize clinical management. However, the manuscript does not sufficiently explore how these imaging findings influenced treatment decisions (e.g., salvage radiation, systemic therapy, or observation) and there is no comparison with conventional imaging.
The study aims to assess whether 18F-DCFPyL PET/CT outperforms conventional imaging in detecting occult BCR, yet:
- No quantitative detection rate comparison between 18F-DCFPyL PET/CT and conventional imaging is provided in the figures.
- The representative images highlight lesions identified by 18F-DCFPyL PET/CT but not by conventional imaging, but without systematic validation, it is difficult to confirm whether these represent true lesions.
- A direct side-by-side analysis of detection rates, ideally stratified by PSA levels or lesion localization, would improve the study’s conclusions.
Author Response
Dear Reviewer, first and foremost, I would like to express my sincerest gratitude for taking the time to review our work and provide your valuable suggestions and comments. Your feedback is greatly appreciated and has undoubtedly contributed to enhancing our manuscript.
I will address each of your comments point by point.
Comments and Suggestions for Authors
This study evaluates the diagnostic performance and clinical impact of 18F-DCFPyL PSMA PET/CT in patients with occult biochemical recurrence of prostate cancer following radical prostatectomy. Given the well-documented limitations of conventional imaging modalities in detecting early recurrence, this work aims to determine how PSMA-based molecular imaging informs therapeutic decision-making. While the study focuses exclusively on 18F-DCFPyL, it does not provide a direct comparison with other PSMA tracers (e.g., 68Ga-PSMA, 18F-Fluciclovine, 11C-Choline PET). Given that multiple PET tracers are available, a clear justification for selecting 18F-DCFPyL over others is necessary.
Point 1: The authors should dedicate a section explicitly discussing the rationale behind this choice, addressing differences in diagnostic accuracy, availability, half-life, resolution, and clinical relevance.
Response and revisions:
Dear Reviewer, regarding to point one, you are right, there is evidence in the literature comparing PET/CT with different PSMA ligands. PSMA ligands differ in terms of the radioisotope used, the underlying radiochemistry, and their biodistribution. Differences in physiological distribution and challenges in image interpretation are well known. However, there is currently no evidence that any specific PSMA radioligand provides superior clinical outcomes over others. Among PSMA ligands, 68Ga-PSMA-11 has the longest history of use and the strongest scientific evidence supporting its efficacy. Currently, 68Ga-PSMA-11, 18F-DCFPyL, and 18F-PSMA-1007 are the most widely used in clinical trials. All of them bind with high affinity to the extracellular domain of PSMA and have demonstrated efficacy in detecting prostate cancer at various stages, with increased uptake correlating with tumor aggressiveness. 18F-DCFPyL has shown a similar lesion detection rate to 68Ga-PSMA-11, without an increase in false positive rates. Additionally, 18F-DCFPyL has demonstrated a higher detection rate compared to 68Ga-PSMA-11, which may be at-tributed to the superior spatial resolution of 18F imaging. Due to their similarities, 68Ga-PSMA-11, 18F-DCFPyL, and 18F-PSMA-1007 are considered a single class of radiotracers and are collectively referred to as PSMA ligands.
We have added a paragraph in the introduction (L 77-81) discussing the different PSMA tracers and have also included a comprehensive description of their advantages and clinical applications, supported by updated references, in the discussion section of our final manuscript (L 353-385)
Point 2: although the discussion briefly acknowledges certain limitations, a dedicated "Limitations and Future Directions" section would provide a clearer understanding for interpreting the study's findings. This section should address: The lack of long-term clinical outcome data (e.g., progression-free survival, overall survival). The absence of comparative analysis between 18F-DCFPyL PET/CT and conventional imaging modalities.
Response and revisions:
Dear Reviewer, we would like to once again thank you for highlighting this important point. We briefly mentioned it in the original manuscript, but we have expanded this information in detail in a separate section of the discussion in our revised manuscript, following your recommendations.
As you rightly point out, there are indeed certain limitations in our study. We started with a relatively small sample (85 patients), which imposed some restrictions on certain statistical analyses. Our study is a prospective investigation with the primary objective of assessing the diagnostic capacity of 18F-DCFPyL PET/CT in patients with biochemical recurrence of prostate cancer following radical prostatectomy. Although it is not the main focus of this study, one of the limitations to highlight is that, at present, we do not have long-term outcome data to evaluate overall survival (OS) and progression-free survival (PFS). Therefore, a second analysis could be proposed for a future publication, where we will evaluate these patients with extended follow-up and include data on PFS and OS.
Regarding your comments on conventional imaging, we are currently working on future publications, including a review of various 18F-labeled PSMA ligands, comparing them directly with conventional imaging techniques. In our current results, no comparative analysis between PET-PSMA and conventional imaging techniques was performed, as the patients selected for the study did not have visible disease on conventional imaging, and the objective was to assess whether they might have disease detectable by metabolic imaging with PET-PSMA. In the future, we plan to include a larger sample in our upcoming publications, as we are actively working to expand our study cohort, focusing on patients with low PSA levels in the context of BCR.
You can find a summary of the limitations in the revised manuscript (L 480-498).
Point 3:
a) The manuscript presents p-values to indicate statistical significance, but the graphical data should be revised for clarity. Specifically: Figures should explicitly indicate which groups show statistically significant differences rather than relying solely on p-values in the text.
Response and revisions:
Dear Reviewer, regarding point three, we would like to express our sincere thanks for your observation. We fully agree that the presentation of the graphical results should be clearer and more explicit. In the revised version of the manuscript, we have added more detailed information on the statistically significant results to facilitate a better interpretation of the figures presented. We have modified several figures to more clearly indicate which groups show statistically significant differences. Additionally, we have included symbols and annotations on the figure legends, and in some cases, directly on the bars of the graphs, to highlight these differences, in addition to the p-values mentioned in the text.
We hope these improvements will provide a clearer visual understanding of the results. We greatly appreciate your suggestion and have worked to make the figures more intuitive and easier to interpret. Thank you once again for your detailed and constructive review.
b)Box plots, confidence intervals, or annotations should be used to improve data interpretation.
Response and revisions:
Regarding this section, we find your suggestion very interesting. When we met with the statisticians, we generated correlation graphs, which I have attached in the word document below, to help illustrate some of the points discussed in the results section of the initial manuscript. However, we initially decided not to include these graphs, as we felt they did not add significant value to the explanation of our results. Now that you have raised this point, we would be happy to consider incorporating them into the final manuscript if you believe they are relevant and could enhance the interpretation of the data in the revised version. We truly appreciate your insightful feedback
Point 4: One of the study's key objectives is to evaluate how positive or negative 18F-DCFPyL PET/CT results optimize clinical management. However, the manuscript does not sufficiently explore how these imaging findings influenced treatment decisions (e.g., salvage radiation, systemic therapy, or observation) and there is no comparison with conventional imaging.
Response and revisions:
We sincerely appreciate all your valuable observations. Regarding the considerations in point four, our primary objective is to evaluate the diagnostic capability of 18F-DCFPyL PSMA PET/CT in patients with BCR of prostate cancer and how its findings whether positive or negative may influence clinical decision-making.
In the results section changes in therapeutic decisions (L 323-340), we have detailed the changes in treatment decisions, comparing the initial treatment intent with the post-PSMA intent for both PET-positive and PET-negative groups. This has also been illustrated schematically in Figure 8 of the revised manuscript. Furthermore, following the recommendation of another reviewer, we have included additional information on the biochemical response after treatment modifications were implemented (L 247-351).
Point 5: The study aims to assess whether 18F-DCFPyL PET/CT outperforms conventional imaging in detecting occult BCR, yet: No quantitative detection rate comparison between 18F-DCFPyL PET/CT and conventional imaging is provided in the figures and the representative images highlight lesions identified by 18F-DCFPyL PET/CT but not by conventional imaging, but without systematic validation, it is difficult to confirm whether these represent true lesions.
Response and revisions:
Regarding these considerations on conventional imaging, we did not perform statistical comparative analyses with conventional imaging because the patients selected for this study were required to have a negative conventional study for N1 or M1 disease. The primary objective was to determine whether PSMA-positive disease could be detected despite a negative conventional imaging result. However, following your suggestions in point two, we have provided a more detailed explanation of this rationale in the limitations and future directions section at the end of the discussion in the revised manuscript (L 481-498).
A direct side-by-side analysis of detection rates, ideally stratified by PSA levels or lesion localization, would improve the study’s conclusions.
Response and revisions:
Finally, we have provided a explanation of this point and reflected it in the section on detection rate according to PSA levels (ng/ml), categorized into groups <0.5, 0.5–1.0, and >1.0 ng/ml, as well as DT-PSA in months, divided into groups <6, 6–12, and >12 months. Additionally, following the recommendation of another reviewer, we have included the detection rate according to ISUP risk groups. All these details have been incorporated into the revised version of the manuscript (L 171-177 and Figure 1).
Once again, thank you for highlighting all these suggestions. Your review has undoubtedly enriched our work, and we sincerely appreciate the time you dedicated to reading our manuscript and providing your valuable contributions.

Round 2
Reviewer 2 Report
Comments and Suggestions for Authors
Thank you for detailed feed-back to reviewer
Point 1: In addition to PSA response you may add overall survival
Point 4. Instead of prevailing approach why not a simple split in three predictive groups?
You may state cohort size was insufficient for logistic regression analyses.
Author Response
Dear Reviewer, first and foremost, I would like to express my sincerest gratitude for taking the time to review our work and provide your valuable suggestions and comments. Your feedback is greatly appreciated and has undoubtedly contributed to enhancing our manuscript.
In the following document, I will address each of your comments point by point.
Comments and Suggestions for Authors
- In addition to PSA response, you may add overall survival
- Instead of prevailing approach why not a simple split in three predictive groups? You may state cohort size was insufficient for logistic regression analyses.
Response and revisions:
Dear Reviewer, we sincerely appreciate your comments in this second round of review. Your previous feedback has undoubtedly enriched our work. Regarding the two minor comments you have raised, our group of authors, along with our statistician, has carefully reviewed them. Our study focuses on the diagnostic power of molecular imaging with 18F-DCFPyL-PSMA in patients with occult biochemical recurrence of prostate cancer. In fact, the primary objective of our study is to measure diagnostic capacity, expressed in terms of the disease detection rate, for the PSA, PSA-DT, and ISUP variables and how these findings could impact decision-making. The objective of our study is not to analyze survival curves. This is not stated in our materials and methods, as it was never the intention of our study.
The research group I lead works with multiple databases, and for this specific article, we used a clinical database where the focus is on diagnostic power and its impact on decision-making based on the study findings. We did not collect data with the aim of analyzing survival curves. In the original manuscript (Volume 1), you will notice that we did not include any data related to what you have suggested. Based on one of the reviewers’ comments in the first round, we examined the PSA kinetics response—more than 50%, less than 50%, and no response—since that reviewer believed it could better complement our results. Everything I am addressing in this response has been incorporated into Version 2 of our manuscript, specifically in the Limitations and Future Directions section. (L 618-635)
Our objective is different for our next two articles, as for those, we have collected comprehensive data for survival curve analysis, which we hope to publish as soon as we complete the analyses.
I would like to once again express my deepest gratitude for your time and the great interest you showed in reviewing our work and providing observations that have undoubtedly improved it.
Reviewer 3 Report
Comments and Suggestions for Authors
The paper has been improved significantly
Author Response
Comments and Suggestions for Authors: The paper has been improved significantlyResponse: Dear Reviewer, Thank you very much for your positive feedback. Your comments have undoubtedly enriched our work.
Please receive our sincere gratitude and warm regards on behalf of the entire team.